# LEARNING TO INTERVENE ON CONCEPT BOTTLENECKS

## ABSTRACT

While deep learning models often lack interpretability, concept bottleneck models (CBMs) provide inherent explanations via their concept representations. Moreover, they allow users to perform interventional interactions on these concepts by updating the concept values and thus correcting the predictive output of the model. Up to this point, these interventions were typically applied to the model just once and then discarded. To rectify this, we present concept bottleneck memory models (CB2Ms), which keep a memory of past interventions. Specifically, CB2Ms leverage a two-fold, differentiable memory to generalize interventions to appropriate novel situations, enabling the model to identify errors and reapply previous interventions. This way, a CB2M learns to automatically improve model performance from a few initially obtained interventions. If no prior human interventions are available, a CB2M can detect potential mistakes of the CBM bottleneck and request targeted interventions. Our experimental evaluations on challenging scenarios like handling distribution shifts and confounded data demonstrate that CB2Ms are able to successfully generalize interventions to unseen data and can indeed identify wrongly inferred concepts. Hence, CB2Ms are a valuable tool for users to provide interactive feedback on CBMs, *e.g.*, by guiding a user's interaction and requiring fewer interventions.

## 1 INTRODUCTION

Deep learning models are often deemed black-box models that make it difficult for human users to understand their decision processes (Adadi & Berrada, 2018; Cambria et al., 2023; Saeed & Omlin, 2023) and interact with them (Schramowski et al., 2020; Teso et al., 2023). To address these issues, one recent branch within explainable artificial intelligence focuses on the potential of concept bottleneck models (CBMs) (Koh et al., 2020; Stammer et al., 2021). These are designed to be partially interpretable and perform inference (such as bird image classification *cf.* Fig. 1 top) by transforming the initial raw input into a set of human-understandable concepts (*e.g.*, wing shape or color) with a bottleneck network. Subsequently, a predictor network provides a final task prediction based on the activation of these concepts. These concept activations serve as an inherent explanation of the model's decision (Teso et al., 2023). Arguably even more valuable, these activations can be used as a means for humans to perform *interventional interactions*, *e.g.*, for querying further explanations (Abid et al., 2022) or correcting concept predictions (Koh et al., 2020).

In fact, a recent surge of research has focused on the benefits of leveraging interactions in AI models in general (Ouyang et al., 2022; Miller, 2019), and also CBMs in particular (Teso et al., 2023). Multiple such approaches focus on leveraging interactions for mitigating errors of the predictor network (Bontempelli et al., 2021; Stammer et al., 2021). So far, little work has focused on mitigating errors of the initial bottleneck network. Moreover, although interventional interactions on a CBM's concept activations are a natural tool for this purpose, they have received little attention since their introduction by Koh et al. (2020). One likely reason for this is that interventions according to (Koh et al., 2020) represent a singular-use tool for updating model performance by adding human-provided concept labels to an increasing number of randomly selected concepts. For sustainably improving a model's performance, however, this approach is inefficient and potentially demands a large number of repetitive user interactions. Providing such repeated feedback has been identified to lead to a loss in focus of human users (Amershi et al., 2014) if not infeasible at all.

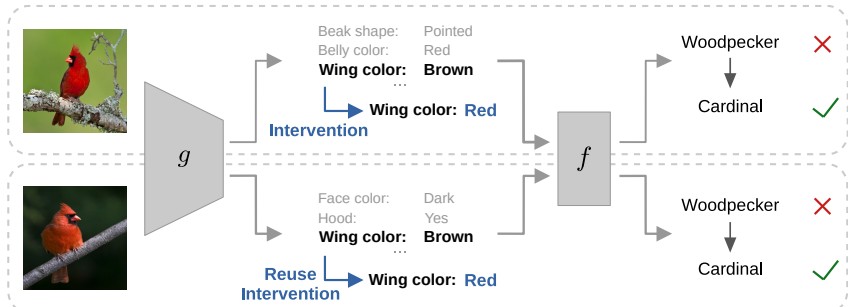

Figure 1: **Reusing a CBM intervention can correct model mistakes for multiple examples.** Top: CBMs generate a human interpretable concept representation via bottleneck ($g$) to solve the final task with a predictor ($f$). Human users can correct these concept predictions via targeted interventions (blue) influencing the final prediction. Bottom: Human interventions hold valuable information reusable in right situations to automatically correct model errors without further human interactions.

In this work, we therefore argue to harvest the rich information present in previously collected interventions in a multi-use approach. Specifically, let us suppose a user corrects a model's inferred concepts through a targeted intervention. In that case, the intervention carries information on where the model did not perform well. As shown in Fig. 1 bottom, this information can be used to improve predictions in similar future situations. In this context, we introduce Concept Bottleneck Memory Models (CB2Ms) as a novel and flexible extension to CBMs. CB2Ms are based on adding a differentiable, two-fold memory of interventions to the CBM architecture, which allows to keep track of previous model mistakes as well as previously applied interventions. This memory enables two important properties for improved interactive concept learning. Specifically, a CB2M can (1) reapply interventions when the base CBM repeats previous mistakes. It thereby automatically corrects these mistakes without the need for additional human feedback. Overall, human feedback may, however, not always be readily available, and obtaining it can be costly. CB2M thus mitigates this issue by (2) its ability to detect potential model mistakes prior to initial human feedback. Its memory module can be used to select data points for human inspection, and thus guide human feedback to where it is really needed. Thus ultimately, CB2Ms allow to overcome the issue of one-time interventions of standard CBMs and enables the model to learn more effectively from targeted human feedback.

We illustrate the full potential of CB2M in our experimental evaluations on several challenging tasks, such as handling distribution shifts and confounding factors across several datasets. In summary, we make the following contributions: (i) We identify the potential of extracting generalizable knowledge from human interventions as a means of correcting concept bottleneck models. (ii) We introduce CB2M, a flexible extension to CBM-like architectures for handling such interactive interventions. (iii) Our experimental evaluations show that CB2Ms can truly learn from interventions by generalizing them to previously unseen examples. (iv) We further show that CB2Ms are also able to detect model mistakes without the need for initial human knowledge and thus allow to query a user for targeted interventions.[1]

## 2 CONCEPT BOTTLENECK MEMORY MODELS (CB2Ms)

Let us first introduce the background notations on CBMs and interventions before presenting CB2Ms to improve interactive concept learning via detecting of model mistakes and generalizing of interventions to novel, unseen examples.

### 2.1 BACKGROUND

A CBM which solves the task of transforming inputs $\mathcal{X}$ to outputs $\mathcal{Y}$ consists of two parts. The bottleneck model $g : x \rightarrow c$ transforms an input $x \in \mathcal{X}$ into its concept representation $c$. Afterwards, the predictor network $f : c \rightarrow y$ uses this representation to generate the final target output $y \in \mathcal{Y}$.

---

[1]code is available publicly at: https://anonymous.4open.science/r/ConceptBottleneckMemoryModels-68F5

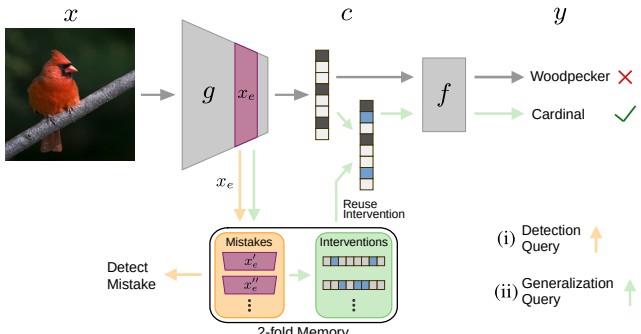

Figure 2: **Overview of CB2M to detect mistakes or generalize interventions.** A vanilla CBM (grey), consisting of bottleneck ($g$) and predictor ($f$), is extended with a two-fold memory (orange and green). The memory compares encodings of new samples to known mistakes to (i) detect model errors or (ii) automatically correct the model via reuse of interventions.

The ground-truth values for $c$ and $y$ are written as $c^*$ and $y^*$, respectively. We refer to overall model (task) accuracy as $\text{Acc}_f$ and to concept accuracy as $\text{Acc}_g$. Human interactions with the concept representations are called interventions. An intervention $i \in \mathcal{I}$ is a set of tuples $i = \{(c'_j, j)|j \in \mathcal{J}_i\}$, with updated concept values $c'_j$ and concept indices $j$. $\mathcal{J}_i$ is the set of all indices for intervention $i$. Applying an intervention to a sample $x$ overwrites the predicted concept values with those of the intervention, which we denote as $x|i$.

As CBMs consist of two processing modules, the bottleneck and predictor networks, errors can occur in either, with different consequences on how to handle these (Bontempelli et al., 2021). If the bottleneck makes an error, this error will most likely also negatively influence the predictor. On the other hand, it is also possible that the predictor makes a wrong final prediction despite having received a correct concept representation. In the latter case, the concept space is either insufficient to solve the task, or the predictor network is susceptible to, *e.g.*, some spurious correlations. Where other works have investigated handling an insufficient concept space through additional (unsupervised) concepts (Sawada & Nakamura, 2022), or correcting a predictor with spurious correlations (Stammer et al., 2021) CB2M on the other hand focuses on mitigating errors that originate from the bottleneck model. This is achieved by utilizing interventions on the concept space. Let us now discuss this in more detail.

## 2.2 CONCEPT BOTTLENECK MEMORY MODELS

Let us now introduce Concept Bottleneck Memory Models (CB2Ms) as a flexible extension to CBM architectures. The bottleneck and predictor networks of the CBM remain unchanged but are extended by a two-fold memory module $\mathcal{M}$ which consists of a *mistake memory* $\mathcal{M}^m$ coupled with an *intervention memory* $\mathcal{M}^i$. The *mistake memory* operates on encodings $x_e$, *i.e.*, the input of the last layer of the bottleneck network. It measures the similarity between two data points $x$ and $x'$, *i.e.*, via the euclidean distance of their encodings, $d(x_e, x'_e) = \|x_e - x'_e\|$. The *intervention memory* directly keeps track of known interventions and associates them to elements of the *mistake memory*, meaning that the memorized intervention $i$ can be used to correct the memorized mistake of $x_e$. We denote an associated encoding and intervention as $\alpha(x_e, i)$.

Overall, this joint memory can be used to detect model mistakes (orange in Fig. 2) or enable automatic reuse of memorized interventions (green in Fig. 2), which we explain in detail in the following paragraphs. Importantly, the character of this memory is independent of the overall CB2M framework. It can be constructed in a differentiable manner, *e.g.*, with neural nearest neighbors (Plötz & Roth, 2018) or, simpler, based on traditional nearest neighbor algorithms.

By extending the vanilla CBM with a memory, CB2M can be used for two distinct tasks (*cf.* Fig. 2): (i) detecting potential model mistakes and (ii) generalizing interventions to new examples. Besides the general advantage of knowing when an AI model has made an incorrect prediction, this knowledge is even more relevant for CBMs as human users can be queried for beneficial interventions in a targeted fashion. Thus, the ability to handle task (i) via CB2M is especially relevant when humans

want to provide interventional feedback to a CBM. Furthermore, after humans have intervened on a CBM, they have, in fact, provided valuable knowledge also for future situations. We claim that this information should not be discarded as in the original work of Koh et al. (2020), but be reused when similar mistakes occur again. This is where task (ii) of CB2M comes into play.

**Detecting Wrongly Classified Instances.** Intuitively, if a data point is similar to other examples where the model made mistakes, the model will more likely repeat these mistakes on the novel data point. Therefore, in CB2Ms the *mistake memory* $M_m$ is utilized to keep track of previous mistakes (*cf.* Alg. 1 in the appendix for pseudo-code). First, the memory is filled with encodings of datapoints, for which the model did not initially generate the correct output and for which the concept accuracy is smaller than a threshold $t_a \in [0, 1]$. This leads to: $\mathcal{M}^m = \{x_e : f(g(x)) \neq y^* \land Acc_g(x) < t_a\}$. For a new unseen instance $\hat{x}$, we then compare its encoding $\hat{x}_e$ with the mistakes in the memory $\mathcal{M}^m$. If we find $k$ mistakes with a distance to $\hat{x}_e$ smaller than $t_d$, we consider a model to be making a known mistake. Formally, we predict a model mistake for a new unseen instance $\hat{x}$ if:

$$\forall j \in \{1, \ldots, k\} : \exists x_{e,j} \in \mathcal{M}^m : d(\hat{x}_e, x_{e,j}) \leq t_d \quad (1)$$

This mistake memory can initially be filled with known model mistakes. Yet, once the CB2M is in use, the memory of mistakes will continuously be updated via interactive feedback, and new encodings will be added. This can constantly improve detection during deployment as corrective interventions can immediately be requested after detecting a potentially misclassified sample.

**Generalization of Interventions.** Next to detecting model errors with the *mistake memory*, we can use both the *mistake memory* and the *intervention memory* jointly to generalize interventions. As initially introduced in (Koh et al., 2020), interventions for correcting predicted concept activations only apply to a single sample. However, we claim that these interventions also contain valuable information for further samples and should thus be reused, thereby reducing the need for additional future human interactions. Intuitively, if an intervention is applicable for one example, it is likely also relevant for similar inputs, at least to a certain degree.

To achieve such intervention generalization from one sample to several, we utilize both parts of the CB2M memory. Specifically, whenever an intervention $i$ is applied to a model, we store it in the *intervention memory* $\mathcal{M}^i$ and keep the encoding of the original input point in the *mistake memory* $\mathcal{M}^m$. We also keep track of corresponding entries $\alpha(x_e, i)$. When the model gets a new sample $\hat{x}$, we next check for similar encodings in the *mistake memory* $\mathcal{M}^m$ according to Eq. 1. Here, we use $k = 1$, considering only the most similar mistake and its intervention. If there is indeed an encoding of a mistake $x_e$ within distance $t_d$ of $\hat{x}_e$, we apply its associated intervention $i$ (with $\alpha(x_e, i)$) to the new data point $\hat{x}$. If there is no similar mistake, we let the model perform its prediction as usual.

The threshold $t_d$ is crucial for intervention generalization, as it directly controls the necessary similarity to reapply memorized interventions. Selecting a suitable value for $t_d$ differs from the mistake prediction as we want to generalize as many interventions as possible under the constraint that the generalized interventions remain valid. To this end, we call an intervention $i$ for a sample $x$ *valid* if the class prediction after intervening is not worse than before. We write this as $valid(x, i) : f(g(x)) = y^* \implies f(g(x|i)) = y^*$. With that, we maximize $t_d$, while keeping:

$$\forall x, x' \in \mathcal{X} : d(x_e, x'_e) \leq t_d \Rightarrow \forall i \in \mathcal{I} : valid(x, i) \Rightarrow valid(x', i) \quad (2)$$

We can also express this in terms of full datasets, where our dataset accuracy after applying interventions should be greater or equal to the accuracy without interventions: $Acc_f(\mathcal{X}|\mathcal{M}) \geq Acc_f(\mathcal{X})$. Here $\mathcal{X}|\mathcal{M}$ is the dataset $\mathcal{X}$ with applied interventions from the memory $\mathcal{M}$:

$$\begin{aligned} \mathcal{X}|\mathcal{M} = &\{x|i : x \in \mathcal{X} : \exists x'_e \in \mathcal{M}^m : \exists i \in \mathcal{M}^i : d(x_e, x'_e) \leq t_d \land \alpha(x'_e, i)\} \\ &\cup \{x : x \in \mathcal{X} : \neg \exists x'_e \in \mathcal{M}^m : d(x_e, x'_e) \leq t_d\} \end{aligned} \quad (3)$$

Thus, we want to find the largest $t_d$ satisfying these constraints. To do that, we can set up the memory $\mathcal{M}$ based on the validation set by adding all model mistakes to $\mathcal{M}^m$ and simulating corresponding interventions with ground-truth labels for $\mathcal{M}^i$. The selection of $t_d$ is then done on the training set. This results in $\mathcal{M}^m = \{x_e : x \in \mathcal{X}_{val} \land f(g(x)) \neq y^*\}$ and $\mathcal{M}^i = \{i : i \in \mathcal{I} \land x_e \in \mathcal{M}^m \land \alpha(x_e, i) \land \forall j \in \mathcal{J}_i : c'_j = c^*_j\}$.

## 3 EXPERIMENTAL EVALUATIONS

To evaluate the potential of CB2Ms in intervention generalization and mistake detection, we perform various evaluations. These include evaluating the ability of CB2Ms to detect similar data points, but also evaluations in the context of unbalanced and confounded data as well as data affected by distribution shifts. Let us first describe the experimental setup.

**Data:** The Caltech-UCSD Birds (CUB) dataset (Wah et al., 2011) consists of roughly 12 000 images of 200 bird classes. We use the data splits provided by Koh et al. (2020), resulting in training, validation, and test sets with 40, 10, and 50% of the total images. Additionally, we add 4 training and validation folds to perform 5-fold validation. Images in the dataset are annotated with 312 concepts (*e.g.*, beak-color:black, beak-color:brown, etc.), which can be grouped into concept groups (one group for all beak-color:_ concepts). We follow the approach of previous work (Koh et al., 2020; Chauhan et al., 2022) and use only concepts that occur for at least 10 classes and then perform majority voting on the concept values for each class. This results in 112 concepts from 28 groups.

We further provide evidence based on the MNIST (LeCun & Cortes, 1998), confounded ColorM-NIST (C-MNIST) (Rieger et al., 2020) and SVHN (Netzer et al., 2011) datasets. For all three, we train the model for the parity MNIST task as in (Mahinpei et al., 2021). Hereby, the digit in the image is considered the concept, and the class label describes whether the digit is even or odd. Furthermore, rather than evaluating on the original MNIST dataset, we focus on an unbalanced version of this task. In this setting, we remove 95% of the training data of one class (for the results in the main paper, the digit "9", for other digits *cf.* App. A.4). We refer to App. A.3 for results on the original MNIST dataset, indicating that current base models yield very high performances and make additional interventions unnecessary. We use the standard train and test splits for these datasets and create validation sets with 20% of the training data. As for CUB, we generate 5 training and validation folds in total. When considering human interventions, we follow the common assumption that humans provide correct concept values as long as the requested concepts are present in the input (*e.g.*, visible in an image).

**Models:** For CUB, we use the same model setup as Koh et al. (2020). For the MNIST variants and SVHN, we follow (Mahinpei et al., 2021). All CBMs are trained with the independent scheme. Further training details can be found in App. A.1. Further training details can be found in App. A.1. We use CB2M as described in Sec. 2.2 to enable the generalization of interventions and detection of model mistakes. CB2M parameters are tuned for generalization and detection separately on the training and validation set (*cf.* App. A.8). For all detection experiments, the memory of CB2M is filled with wrongly classified instances of the validation set according to the parameters. For generalization experiments, we simulate human interventions on the validation set and use CB2M to generalize them to the test set.

**Metrics:** We use both concept and class accuracy of the underlying CBM (with and without CB2M) to observe improvements in the final task and to investigate the intermediate concept representation. We evaluate the detection of model mistakes using the area under the receiver operating characteristic (AUROC) and the area under precision-recall curve (AUPR), in line with related work (Ramalho & Miranda, 2019). To observe how interventions improve model performance, we propose normalized relative improvement (NRI), which measures improvement independent of baseline values. NRI measures the percentage of the maximum possible improvement in class accuracy achieved as $\text{NRI} = \Delta/\Delta_{\max} = (\text{Acc}_f - \text{Acc}_{f,\text{base}})/(\text{Acc}_{f,\max} - \text{Acc}_{f,\text{base}})$. Where $\text{Acc}_f$ ($\text{Acc}_{f,\text{base}}$) refers to the model accuracy after (before) applying interventions and $\text{Acc}_{f,\max}$ is the maximum possible accuracy to achieve through interventions, estimated, *e.g.*, by the accuracy of the predictor given ground-truth concept information on the validation set.

### 3.1 RESULTS

**Beyond One-Time Interventions.** First, we analyze how well CB2M generalizes interventions to unseen data points. If a standard CBM receives a new input similar to a previous datapoint with a corresponding intervention, that intervention is not further used. CB2M, on the other hand, allows the reuse of information provided in previous interventions. As CB2M has access to more information than the base CBM, we also compare it against a CBM, which is finetuned on the data used to generate interventions for CB2M for different number of finetuning steps (until convergence).

Table 1: **CB2M generalizes interventions to unseen data points.** Top: Performance of CBM, finetuned CBMs and CB2M on the full dataset. Generalizing interventions with CB2M improves upon the base CBM on all cases. CBM (ft) achieves higher class accuracy in two cases, but does not provide any improvements on Parity MNIST (unbalanced) Bottom: Particularly, CB2M identifies incorrect instances and generalizes suitable interventions to them. (Best values bold, average and standard deviation over augmented test set versions CUB (Aug.) or 5 runs (other)).

| Dataset | Set. | Concept Acc. ($\uparrow$) | | | Class Acc. ($\uparrow$) | | |
| | | CBM | CBM (ft) | CB2M | CBM | CBM (ft) | CB2M |
| --- | --- | --- | --- | --- | --- | --- | --- |
| CUB (Aug.) | Full | $94.7 \pm 0.6$ | $96.2 \pm 0.3$ | $\mathbf{98.7} \pm 3.5$ | $64.8 \pm 2.7$ | $\mathbf{74.7} \pm 1.8$ | $69.1 \pm 5.5$ |
| P MNIST (ub) | Full | $97.5 \pm 0.2$ | $97.9 \pm 0.1$ | $\mathbf{98.0} \pm 0.3$ | $91.2 \pm 0.1$ | $91.8 \pm 0.4$ | $\mathbf{94.0} \pm 1.2$ |
| P C-MNIST | Full | $87.1 \pm 0.0$ | $\mathbf{95.0} \pm 0.1$ | $88.4 \pm 0.4$ | $68.6 \pm 0.3$ | $\mathbf{88.1} \pm 0.8$ | $74.9 \pm 2.1$ |
| CUB (Aug.) | Id | $86.4 \pm 2.7$ | - | $\mathbf{99.0} \pm 0.7$ | $5.0 \pm 1.7$ | - | $\mathbf{88.7} \pm 5.4$ |
| P MNIST (ub) | Id | $85.3 \pm 2.6$ | - | $\mathbf{98.7} \pm 0.4$ | $22.5 \pm 5.7$ | - | $\mathbf{93.7} \pm 1.9$ |
| P C-MNIST | Id | $82.2 \pm 0.6$ | - | $\mathbf{95.5} \pm 1.2$ | $20.1 \pm 7.1$ | - | $\mathbf{85.9} \pm 4.7$ |

Specifically, CBM (ft) was finetuned for 10 epochs on CUB and 5 epochs on the Parity MNIST variants. To evaluate the generalization of CB2M to datapoints similar to the intervened samples, we provide results on a modified version of the CUB dataset: CUB (Aug.). We augment the dataset with color jitter, blurring, blackout, as well as salt&pepper, and speckles noise, to obtain images that correspond to similarly challenging natural image recording conditions, *e.g.*, a change in lighting. We then fill CB2M with simulated human interventions on the unmodified test set and generalize them to the novel augmented test set version. The results of these evaluations in Tab. 1 show that indeed CB2M substantially improves upon the base CBM on instances identified (Id) for intervention generalization, and consequently also on the full data set (Full)[2]. (*cf.* App. A.6 for further information on false positive/negative rates and App. A.5 regarding the validation set size).

Next, we evaluate CB2M under more challenging settings, training with highly unbalanced or confounded data. As seen in Tab. 1 the base CBM struggles to learn the underrepresented digit in the unbalanced Parity MNIST dataset. On the confounded Parity C-MNIST dataset[3] the CBM is strongly influenced by the confounding factor which negatively impacts the bottleneck performance during test time. By generalizing from few human interventions, CB2Ms can substantially improve performance compared to the vanilla CBM on both datasets. Specifically, the reapplied interventions reach a concept accuracy close to 100%, showing that the interventions successfully correct the bottleneck errors. Furthermore, correcting the concept representation on those instances that were identified for reapplied interventions substantially boosts the class accuracy on these instances. Overall, these results show that CB2Ms are very successful in generalizing interventions. This holds not only for naturally similar inputs, but also for scenarios like unbalanced and confounded data.

We note that, while CB2M shows superior performances than CBM, extended finetuning (CBM (ft)) does provide notable improvements particularly for Parity C-MNIST both in terms of concept and class accuracy and slight improvements in class accuracy for CUB (Aug.). This effect is however not observed for Parity MNIST. Moreover, next to the raw performance, there are other aspects to consider when comparing CB2M with finetuning the base CBM. Particularly, finetuning a model can be costly, even more so if the model is very large. This can render repeated finetuning on interventional data during deployment infeasible. The memory of CB2M on the other hand can be directly adapted without additional optimization costs, but can result in slightly higher inference costs (*cf.* App. A.1). Moreover, CB2M can provide potential benefits in an online setting over vanilla fine-tuning, when the model should be continuously updated with new interventional data., *e.g.*, via explicitly memorizing previous mistakes. In general, finetuning removes all other benefits of having an accessible memory in the context of interpretablity and interactability. Specifically, it is difficult to remove already applied interventions from the finetuned model, if it turns out the interventions were incorrect. Inspecting the representation of the finetuned model is also difficult, where in CB2M a user can simply inspect the model's memory. Overall, our results and considerations suggest that parameter finetuning and CB2M can be viewed as complementary approaches for model revisions via interventions.

---

[2]This distinction is not relevant for CBM (ft) as it does not explicitly identify model mistakes.

[3]For this dataset, we assume that we have access to some human interventions on unconfounded data.

Table 2: **CB2M detects wrongly classified instances.** AUROC and AUPR values on the test set. For the confounded Parity C-MNIST, CB2M can even achieve substantially better detection than the baselines. (Best values bold, average and standard deviations over 5 runs.)

| Dataset | Confounded | Metric | Random | Softmax | CB2M |
|---|---|---|---|---|---|
| CUB | No | AUROC ($\uparrow$) | $51.1 \pm 0.7$ | $83.7 \pm 1.1$ | $\mathbf{84.8} \pm 0.7$ |
| | | AUPR ($\uparrow$) | $77.3 \pm 0.4$ | $94.0 \pm 0.6$ | $\mathbf{94.6} \pm 0.3$ |
| CUB (conf) | Yes | AUROC ($\uparrow$) | $49.4 \pm 0.8$ | $77.4 \pm 1.1$ | $\mathbf{85.1} \pm 0.5$ |
| | | AUPR ($\uparrow$) | $76.7 \pm 0.4$ | $91.5 \pm 0.7$ | $\mathbf{94.6} \pm 0.3$ |
| Parity MNIST (unbalanced) | No | AUROC ($\uparrow$) | $50.5 \pm 0.1$ | $\mathbf{90.7} \pm 1.7$ | $88.7 \pm 0.4$ |
| | | AUPR ($\uparrow$) | $91.2 \pm 0.1$ | $\mathbf{98.8} \pm 0.3$ | $98.5 \pm 0.1$ |
| Parity C-MNIST | Yes | AUROC ($\uparrow$) | $50.3 \pm 0.7$ | $65.7 \pm 0.3$ | $\mathbf{83.4} \pm 0.8$ |
| | | AUPR ($\uparrow$) | $69.0 \pm 0.6$ | $79.8 \pm 0.3$ | $\mathbf{91.5} \pm 0.4$ |

Table 3: **Interventions based on CB2M detection successfully improve model performance.** NRI of interventions on identified instances and full test set. As expected, interventions improve performance on identified instances for all methods. More importantly, using CB2M leads to considerably larger improvements on the full dataset. (Best values bold, standard deviations over 5 runs.)

| Setting | Random | Softmax | CB2M |
|---|---|---|---|
| | CUB | | |
| Identified | $95.4 \pm 0.6$ | $\mathbf{96.3} \pm 0.6$ | $95.9 \pm 0.5$ |
| Full Set | $34.3 \pm 5.7$ | $70.1 \pm 3.1$ | $\mathbf{75.5} \pm 4.5$ |
| | Parity MNIST (unbalanced) | | |
| Identified | $\mathbf{100.0} \pm 0.0$ | $\mathbf{100.0} \pm 0.0$ | $\mathbf{100.0} \pm 0.0$ |
| Full Set | $13.2 \pm 4.2$ | $62.1 \pm 4.9$ | $\mathbf{69.6} \pm 4.1$ |
| | Parity C-MNIST | | |
| Identified | $\mathbf{100.0} \pm 0.0$ | $\mathbf{100.0} \pm 0.0$ | $\mathbf{100.0} \pm 0.0$ |
| Full Set | $60.0 \pm 9.8$ | $87.3 \pm 0.8$ | $\mathbf{89.7} \pm 6.1$ |

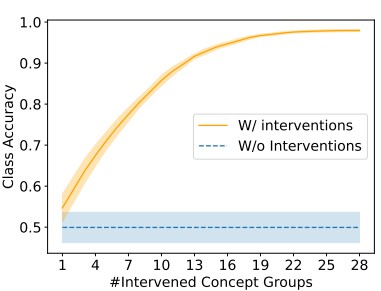

Figure 3: **Less is enough: Intervening on a subset of all concepts already yields large improvements.** CB2Ms can be combined with methods which select subsets of concepts for interventions (here ECTP) (Shin et al., 2023). (Mean and std over 5 runs)

**Asking for Interventions.** Next, we go from the generalization of provided interventions to the second use-case of CB2Ms, namely for detecting model mistakes prior to human feedback. For this, we compare CB2M to two baselines. The *random* baseline for mistake detection simply marks random samples as mistakes. In contrast, *softmax* based detection of mistakes uses the softmax probability of the strongest activated class as a proxy to predict whether the model made a mistake (Hendrycks & Gimpel, 2017). Where the *softmax* baseline uses information from the end of the model, *i.e.*, after the predictor network, CB2Ms estimate model errors only based on the bottleneck network. While detecting mistakes of the whole model covers all potential model errors (*i.e.*, bottleneck and predictor), we hypothesize that detecting mistakes of the bottleneck network directly via CB2M is more suitable for interventions, as they are tied to the bottleneck network. We compare CB2M to the baselines on CUB and the Parity MNIST (unbalanced) datasets. Additionally, we evaluate the detection on Parity C-MNIST and the confounded version of CUB: CUB (conf), where the methods have access to a small number of unconfounded data points. Our results in Tab. 2 indicate that the mistake detection of CB2Ms performs on par with *softmax* on CUB and Parity MNIST (unbalanced). But particularly mistake detection via CB2Ms is superior to *softmax* on the two confounded datasets, as it is able to make better use of the small number of unconfounded samples.

**Improving detected mistakes.** Next, we show that once model mistakes have been detected, human interventions provide a straightforward way to improve a model via the detected mistakes. Specifically, for this we evaluate the effect of interventions on model performance when these are applied on the previously detected mistakes of CB2Ms. In Tab. 3, we report the normalized relative improvement (NRI) on the test set to evaluate the improvement due to interventions that were applied to previously detected mistakes. We observe that both for CUB and Parity MNIST (unbalanced),

Table 4: **CB2M generalization under distribution shift.** The CBM is trained on Parity MNIST and evaluated on SVHN. Despite the low base model performance, CB2M can still generalize human interventions on SVHN. (Best values bold, standard deviations over 5 runs.)

| Setting | Concept Acc. (↑) | | Class Acc. (↑) | |
|---|---|---|---|---|
| | CBM | CB2M | CBM | CB2M |
| Identified | $63.1 \pm 1.2$ | $\mathbf{87.3} \pm 0.1$ | $39.9 \pm 0.3$ | $\mathbf{60.8} \pm 0.4$ |
| Full set | $68.0 \pm 0.9$ | $\mathbf{75.3} \pm 0.4$ | $51.0 \pm 0.1$ | $\mathbf{57.3} \pm 0.2$ |

interventions can improve model performance on detected mistakes, resulting in (close to) 100% test accuracy. This results in similar NRIs for all methods on the identified instances. More important, however, is the effect observed on the full dataset. Here, we can see that interventions after random selection only have a small effect. Interventions applied after the softmax baseline and CB2M yield substantially larger improvements, though, overall the results hint that CB2Ms can detect mistakes more suitable for interventions.

**Interventions on subsets of concepts.** Often, intervening on a few concepts is already sufficient because they carry most of the relevant information. As human interactions are expensive, we want to only ask for interventions on the relevant concepts. As shown in Shin et al. (2023) and Chauhan et al. (2022), selecting specific concepts for interventions can greatly reduce the required human interactions. To show that this holds also in the context of CBMs, in Fig. 3, we exemplarily combine CB2M with the concept subset selection method ECTP (Shin et al., 2023). This figure shows the increase in performance when applying interventions after CB2M detection for a progressive number of concepts. One can observe that interventions on a few concept groups (10) already yield a large portion of the maximum improvement (60%). Applying interventions beyond 19 concept groups barely shows further improvements. This highlights that we do not necessarily need interventions on all concepts to achieve benefits of CB2Ms, but they can be combined with existing methods which perform concept selection for individual samples.

**Generalization under Distribution Shift.** Lastly, we want to evaluate the benefits of CB2M when the base CBM is affected by a distribution shift. To that end, we first train a CBM on Parity MNIST and then evaluate it on Parity SVHN. As seen in Tab. 4, the base model does not perform well under the shift, with a class accuracy barely over 50% (which is equal to random guessing). Nevertheless, we observe that if we add human-generated interventions to CB2M, we can greatly improve the model performance despite the distribution shift, indicating the great potential of CB2Ms also in other learning settings such as online learning.

**Limitations.** With CB2Ms, we leverage human feedback to improve upon CBMs. To this end, it is assumed that the feedback provided by humans is correct. This is a common assumption in work on CBMs (Koh et al., 2020; Chauhan et al., 2022) and (inter)active learning in general (Settles, 2009; Berg et al., 2019). However, despite a human's ability (*e.g.*, sufficient expertise) to provide correct feedback, a user with malicious intentions could actively provide wrong feedback. This has to be considered when incorporating human feedback, *i.e.*, also in the context of CB2M. Recent work has begun tackling this issue *e.g.*, in the context of explanatory interactive learning (Friedrich et al., 2023), toxic language (Ju et al., 2022) and specifically concept-based AI systems (Collins et al., 2023). Moreover, inefficient search and memory storage can affect the usability of CB2Ms in large-scale practical settings. Lastly, a more fundamental issue of CBMs is that a high sample-variance in terms of concept encodings can potentially lead to a higher amount of required interventions.

## 4 RELATED WORK

**Concept Bottleneck Models.** Concept bottleneck models as a general network architecture were popularized recently by Koh et al. (2020). The two staged model first computes intermediate concept representations before generating the final task output. Since their introduction, various extensions and variations of the standard CBM architecture were introduced. To depend less on supervised concept information, CBM-AUC (Sawada & Nakamura, 2022) combine explicit concept supervision with unsupervised concept learning. Similarly, PostHoc CBMs (Yüksekgönül et al., 2022) and label-free CBMs (Oikarinen et al., 2023) encompass concepts from concept libraries (*e.g.*, with

CAV (Kim et al., 2018)) to require less concept supervision and Stammer et al. (2022) learn concepts directly with weak supervision based on discretizing prototype representations. Other extensions to CBMs aim to mitigate concept leakage (Margeloiu et al., 2021), ensuring the inherent interpretability of CBMs. Examples are GlanceNets (Marconato et al., 2022) and CEM (Zarlenga et al., 2022). In another line of work, Lockhart et al. (2022) enable CBMs to drop the concept predictions if not enough knowledge is available. This large variety of CBM-like architectures makes the flexibility of our presented CB2M desirable. The only requirements to combine CB2M with other CBM architectures are access to the model encodings and the ability to apply interventions.

As a two-stage model, CBMs have many advantages compared to standard deep models, but their structure can make error analysis also more difficult (Marconato et al., 2023). Due to separate processing of inputs via the bottleneck and predictor networks, error sources also have to be tackled individually (Bontempelli et al., 2021). Where several previous works have tackled mitigating errors in the predictor network (Sawada & Nakamura, 2022; Stammer et al., 2021; Teso et al., 2023), interventions are a tool to tackle bottleneck errors. However, the initial introduction of interventions applies them to random concepts for all samples (Koh et al., 2020), which is no efficient use of human interactions. Since then, Shin et al. (2023) proposed several heuristics to order concepts for intervention and SIUL (Sheth et al., 2022) uses Monte Carlo Dropout to estimate concept uncertainty for the same purpose. Interactive CBMs (Chauhan et al., 2022) extend the idea even further by providing a policy to optimize concept selection under consideration of intervention costs. Still, all these works only consider ordering of concepts for interventions. With CB2M, we provide a mechanism to handle bottleneck errors via interventions specifically when they occur. And even more importantly, CB2M allows interventions to have more than a one-time effect.

**Uncertainty Estimation for Error Detection.** One use case of CB2Ms is to detect potential model mistakes (which can then be improved via interventions). Detecting data points where models perform poorly is often touched upon in research on uncertainty estimation. While the construction of uncertainty-aware networks provides benefits in terms of mistake detection (Gawlikowski et al., 2021), our work is more related to methods without particular assumptions on the model architecture. This ensures that CB2M can be combined with different CBM architectures. A popular approach to detect model mistakes is using softmax probabilities of the most likely class (Hendrycks & Gimpel, 2017). However, these methods are not specifically tailored to CBMs. They are able to detect model mistakes in general, while CB2M can specifically detect mistakes related to the bottleneck, which can be corrected via interventions. In contrast, NUC (Ramalho & Miranda, 2019) learn a neural network on top of a KNN of latent model representations to predict uncertainty. We do not learn a neural network on top of similarity information, thus keeping our technique simpler and more flexible *e.g.*, when novel details about model mistakes arrive at model deployment.

## 5 CONCLUSION

In this work, we have introduced CB2M, a flexible extension to CBM models. We have shown that the two-fold memory of CB2Ms can be used to generalize interventions to previously unseen datapoints, thereby overcoming the issue of current one-time intervention approaches without the necessity of further human interactions. Furthermore, we have demonstrated that CB2Ms can be utilized to detect model mistakes prior to any human interactions, allowing humans to efficiently provide interventional feedback in a targeted manner, based on model-identified mistakes. Overall, our experimental evidence on several tasks and datasets shows that CB2Ms can be used to greatly improve intervention effectiveness for efficient interactive concept learning.

A promising avenue for future enhancements of CB2M is instantiating the memory in a differentiable way which would allow to learn parameters directly instead of relying on heuristics. Aggregating interventions from multiple similar mistakes, *i.e.*, using $k > 1$ for generalization could increase robustness of reapplied interventions, while aggregation them in the memory via prototypes could keep the memory small and better understandable. It is further important to investigate the potential use-case of CB2Ms in the context of continual learning (*e.g.*, concerning robustness to catastrophic forgetting) and the potential of combining CB2M with important previous works *e.g.*, (Aljundi et al., 2019). Finally, an interesting future direction is the combination of CB2M with other concept-based models, for example CEM (Zarlenga et al., 2022), post-hoc CBMs (Yüksekgönül et al., 2022) or even tabular CBMs (Zarlenga et al., 2023).

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

# A APPENDIX

## A.1 ADDITIONAL EXPERIMENTAL DETAILS

**Model Training:** For CUB, we use the same model setup as (Koh et al., 2020), instantiating the bottleneck model with the Inception-v3 architecture (Szegedy et al., 2016) and the predictor network with a simple multi-layer perceptron (MLP). On the Parity MNIST, SVHN, and C-MNIST datasets, we used an MLP both for the bottleneck and predictor networks. The bottleneck is a two-layer MLP with a hidden dimension of 120 and ReLU activation functions, while the predictor is a single-layer MLP. The bottlenecks are trained using the specific dataset's respective training and validation sets. Notably, for the Parity MNIST (unbalanced), the training unbalance is not present in the validation data. For the generalization and mistake detection experiments on C-MNIST, the human-provided interventions are from the unconfounded data, which is 10% of the original C-MNIST test dataset, which was neither used for training nor evaluation. Evaluation is done on the remainder of the test set. For the the distribution shift experiment of SVHN, we used a validation set of 20% of the training set as base for the interventions.

**Assumptions About Human Feedback.** With CB2Ms, we leverage human feedback to improve upon CBMs. To this end, it is assumed that the feedback provided by humans is correct. This is a common assumption in work on CBMs (Koh et al., 2020; Chauhan et al., 2022) and (inter)active learning in general (Settles, 2009; Berg et al., 2019). For humans, it is often easier to provide concept information than to provide information on the complete task. For example, when considering bird species classification *cf.* Fig. 1, it is easier to identify the bird's color than its species. This phenomenon occurs when concepts are "low-level" and human-understandable. In other domains, such as the medical one, providing correct concept labels may require expert domain knowledge, but it is still possible and easier to infer concept labels than class labels.

**Size of the Memory Module.** When more and more interventions get added to the memory, this increases the evaluation time to reapply interventions. However, as various other work in the context of knowledge-based question answering has shown (Borgeaud et al., 2022; Lewis et al., 2020), it is possible to scale neighbor-based retrievers to millions of data points. In particular, approximate nearest neighbor inference (*e.g.*, FAISS (Johnson et al., 2021)), allows to scale NNs. Furthermore, it is unlikely that the memory of CB2M would reach such dimensions, as it is filled based on human interactions. Therefore we argue that even if the size of the memory has an impact on the evaluation runtime, this is not a major drawback. Nevertheless, a large memory can cause certain drawbacks, as *e.g.*, reduced interpretability of the memory. Therefore, we think that methods to reduce the number of elements in the memory (*e.g.*, prototypes), could be a promising avenue for future research.

## A.2 ALGORITHMS FOR INTERVENTION GENERALIZATION AND MISTAKE DETECTION

For reference, we present algorithms with pseudo code for mistake detection (Alg. 1) and intervention generalization (Alg. 2).

---

**Algorithm 1 Detection of Model Mistakes.** Given: Parameters $t_d$, $t_a$ and $k$, data set for memory setup (e.g. validation set) $\mathcal{X}_{val}$ and a CBM with bottleneck $f$ and predictor $g$.

---
1: **Memory setup:** $\mathcal{M}^m \leftarrow \{x_e : x \in \mathcal{X}_{val} \wedge f(g(x)) \neq y^* \wedge Acc_g(x) < t_a\}$
2: $\hat{x} \leftarrow$ `New unseen instance`; $j \leftarrow 0$
3: **for** $m \in \mathcal{M}^m$ **do**
4:     **if** $d(\hat{x}_e, m) \leq t_d$ **then**
5:         $j \leftarrow j + 1$
6:     **end if**
7: **end for**
8: **if** $j \geq k$ **then**
9:     **return** `Mistake`
10: **else**
11:     **return** `No mistake`
12: **end if**

---

---

**Algorithm 2 Generalize Interventions to Unseen Images** Given: CBM with bottleneck $g$ and predictor $f$, threshold parameter $t_d$ and a memory $\mathcal{M} = (\mathcal{M}^m, \mathcal{M}^i)$ of reference mistakes with respective interventions.

---

1: $\hat{x} \leftarrow$ New unseen instance
2: **Obtain** $\hat{x}_e$ **through** $g$
3: **find** $x' \in \mathcal{M}^m$ with minimal $d(\hat{x}_e, x'_e)$
4: **if** $d(\hat{x}_e, x'_e) < t_d$ **then**
5:     **if** $\exists i \in \mathcal{M}^i : \alpha(x'_e, i)$ **then**
6:         $x \leftarrow x|i$
7:     **end if**
8: **end if**
9: **Model Output:** $y = f(x)$

---

Table 5: **Detection of model mistakes on Parity MNIST**. For mistake detection on models with a low error rate (with errors being outliers close to the decision boundaries), CB2M performs worse than softmax. (Best values bold, standard deviations over 5 runs.)

| | Random | Softmax | CB2M |
|---|---|---|---|
| AUROC ($\uparrow$) | $49.0 \pm 0.4$ | $\mathbf{93.3} \pm 0.2$ | $64.6 \pm 1.0$ |
| AUPR ($\uparrow$) | $97.4 \pm 0.0$ | $\mathbf{99.8} \pm 0.0$ | $98.8 \pm 0.1$ |

## A.3 Results on Parity MNIST

For reference, we provide results when applying CB2M to Parity MNIST. The performance of the base CBM on this task is already pretty good, as it achieves a concept accuracy of $98.9\%$ and a class accuracy of $97.7\%$. The few errors that the model makes are due to singular outliers. As discussed in Sec. 5, the CB2M performs well when the model is subject to some kind of systematic error, *e.g.*, when the model is subject to a shift in data distribution or due to data imbalance at training time. When model mistakes are just a few individual examples, which are getting confused with different classes, CB2M does not perform as well (Tab. 5, 6). As the base CBM performance is already good, further intervention generalization is not suitable, as the remaining model mistakes are not similar to each other (Tab. 7). Further adjustments like including positive examples in the memory or using an explicit view on mistake density could potentially improve results in these situations.

## A.4 Further Results on Parity MNIST (unbalanced)

The unbalanced version of Parity MNIST is generated by dropping $95\%$ of the training data of one class. In the main paper, we exemplarily showed the results when removing digit $9$. In Tab. 8, we show the average results for all other digits. The base mode does not capture the training imbalance properly in three cases, resulting in larger standard deviations for all results.

## A.5 Evaluations on Varying Validation Set Sizes

In Fig. 4 we provide additional results of the generalizaton experiment based on different subset sizes of the validation set. Specifically, we present the concept and class accuracies for CB2M models that were provided with 25%, 50%, 75% or 100% of the validation set. These evaluations

Table 6: **Interventions after detection on Parity MNIST.** NRI on identified instances and full set. Interventions successfully improve identified instances. However, worse detection than softmax results in smaller improvement via CB2M. (Best values bold, standard deviations over 5 runs.)

| Setting | Random | Softmax | CB2M |
|---|---|---|---|
| Identified | $\mathbf{100.0} \pm 0.0$ | $\mathbf{100.0} \pm 0.0$ | $\mathbf{100.0} \pm 0.0$ |
| Full Set | $1.6 \pm 0.7$ | $\mathbf{57.6} \pm 1.5$ | $5.9 \pm 2.9$ |

Table 7: **Generalization of CB2M does not impact results on Parity MNIST**. As model mistakes are not similar to each other, no instances have been identified for intervention generalization, therefore applying CB2M does not impact model performance. (Best values bold, standard deviations over 5 runs.)

|  | Setting | CBM | CB2M |
|---|---|---|---|
| Concept Acc. (↑) | Identified | ∅ | ∅ |
|  | Full set | $\mathbf{98.9} \pm 0.0$ | $\mathbf{98.9} \pm 0.0$ |
| Class Acc. (↑) | Identified | ∅ | ∅ |
|  | Full set | $\mathbf{97.7} \pm 0.0$ | $\mathbf{97.7} \pm 0.0$ |

Table 8: **Further results on Partiy MNIST (unbalanced).** Results of all main experiments for all versions of the Parity MNIST (unbalanced) dataset (where the digits 0 to 8 where the underrepresented digits respectively). (Average and standard deviation over unbalance with digits 0 to 8.)

| | Mistake Detection | | |
|---|---|---|---|
|  | Random | Softmax | CB2M |
| AUROC (↑) | $49.5 \pm 1.0$ | $\mathbf{91.2} \pm 7.2$ | $83.8 \pm 10.77$ |
| AUPR (↑) | $92.8 \pm 3.5$ | $\mathbf{99.3} \pm 0.4$ | $98.9 \pm 0.3$ |
| | Performance after Interventions (NRI) | | |
| Setting | Random | Softmax | CB2M |
| Identified | $\mathbf{100.0} \pm 0.0$ | $\mathbf{100.0} \pm 0.0$ | $\mathbf{100.0} \pm 0.0$ |
| Full Set | $24.3 \pm 21.3$ | $70.7 \pm 13.8$ | $\mathbf{75.6} \pm 14.6$ |
| | Generalization of Interventions | | |
|  | Setting | CBM | CB2M |
| Concept Acc. (↑) | Identified | $91.8 \pm 2.1$ | $\mathbf{97.4} \pm 2.3$ |
|  | Full Set | $98.4 \pm 0.0$ | $\mathbf{98.6} \pm 0.3$ |
| Class Acc. (↑) | Identified | $37.2 \pm 26.4$ | $\mathbf{87.1} \pm 11.4$ |
|  | Full Set | $92.9 \pm 3.5$ | $\mathbf{95.0} \pm 2.9$ |

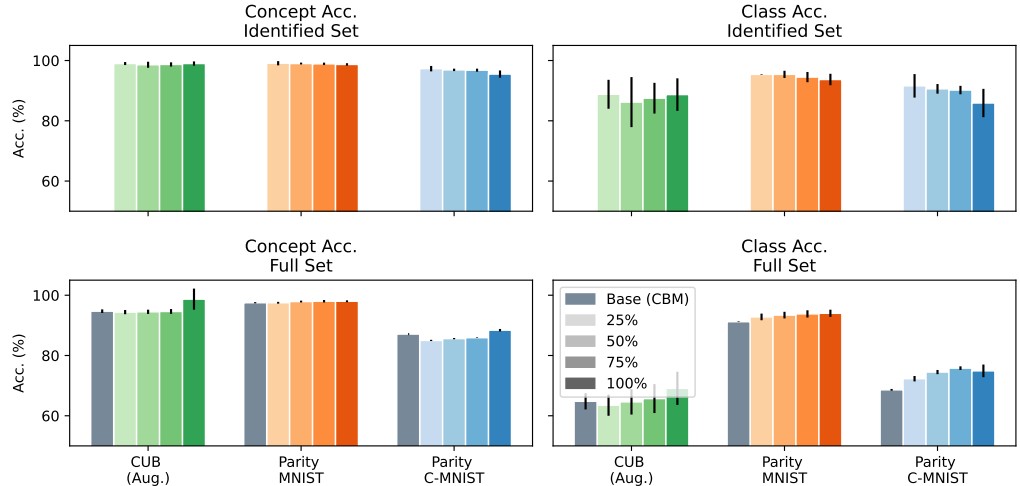

Figure 4: Ablation evaluation on the effect of the validation size on the concept and class accuracy for the identified and full set. The CB2M models were were provided with 25%, 50%, 75% or 100% of the validation set. We present the baseline CBM results for comparisons.

Table 9: Number of generalized interventions for the different datasets. For SVHN, the number of model mistakes is considerably larger, therefore there are more possible generalizations. (Average and standard deviations over 5 runs.)

| Dataset | Number of Intervention Generalizations |
|---|---|
| CUB | $289.4 \pm 215.5$ |
| Parity MNIST (unbalanced) | $416.2 \pm 206.5$ |
| Parity C-MNIST | $913.4 \pm 342.8$ |
| Parity MNIST to SVHN | $7809 \pm 512$ |

were performed on the augmented version of CUB (CUB (Aug.)), the parity MNIST and the parity C-MNIST data sets. We observe that for CUB (Aug.) the effect scales roughly with the validation set size, which is to be expected, as interventions are mostly reapplied to augmented versions of the same sample. For the other datasets, the effect on the full set better, the results between 50% and 100% of the validation set size are relatively similar, meaning that in situations where the CB2M is used to prevent a systematic error of the base CBM, we do not need as many human interventions.

## A.6 FURTHER DETAILS ON GENERALIZATION RESULTS

In Sec. 3.1, we show the generalization capabilities of CB2Ms on various datasets. To further detail these results, the number of generalized interventions is presented in Tab. 9. This describes to how many unseen examples the human interventions have been generalized. The standard deviation is generally relatively large, especially for the CUB dataset. This is most likely due to two reasons. First, the threshold parameter $t_d$ was selected the same for all augmentations, possibly not optimal for all augmented versions. Additionally, the two augmentations salt&pepper and speckles noise have a disruptive effect on the model encodings, causing substantially fewer samples to be selected for intervention generalization than for the other augmented versions. The number of generalized interventions for the parity MNIST to SVHN dataset is considerably larger, as this dataset has more datapoints, and the model makes more mistakes after the distribution shift.

To further investigate the effect of finetuning on the interventional data, we provide more results in Tab. 10. We compare finetuning for a short amount of time (1 epoch), to extended finetuning (5 epochs for MNIST variants and 10 epochs on CUB (Aug.)). One can observe that longer finetuning is necessary to obtain its benefits, as short finetuning does not surpass the performance of CB2M.

Table 10: Finetuning a CBM on the validation set. *Short* and *long* refer to the number of finetuning steps, i.e. 1 epoch for short and 10 epochs for finetuning on CUB and 5 epochs for finetuning on the MNIST versions. (Average and standard deviations over 5 runs.)

| Dataset | Concept Acc. (↑) | | Class Acc. (↑) | |
|---|---|---|---|---|
| | CBM (short) | CBM (long) | CBM (short) | CBM (long) |
| CUB | $95.2 \pm 0.1$ | $96.28 \pm 0.3$ | $67.38 \pm 1.9$ | $74.66 \pm 1.8$ |
| Parity MNIST (unbalanced) | $98.2 \pm 0.1$ | $97.9 \pm 0.1$ | $91.77 \pm 0.5$ | $91.78 \pm 0.4$ |
| Parity C-MNIST | $89.9 \pm 0.1$ | $95.0 \pm 0.1$ | $70.6 \pm 0.4$ | $88.1 \pm 0.8$ |

Table 11: False positive rates and false negative rates for the identification of samples to reapply an intervention (Tab. 1).

| Dataset | FPR | FNR |
|---|---|---|
| CUB | $0.84 \pm 0.43$ | $86.94 \pm 8.76$ |
| Parity MNIST (unbalanced) | $1.14 \pm 0.80$ | $64.39 \pm 15.4$ |
| Parity C-MNIST | $3.23 \pm 0.17$ | $73.25 \pm 1.07$ |

Additionally, for Parity MNIST (unbalanced), finetuning independent of the number of steps does not provide noticable improvements.

In Tab. 11, we provide the false positive rate (FPR) and false negative rate (FNR) for all generalization experiments of Tab. 1. The FPR is the fraction of negative samples (no mistake of the CBM), which gets a reapplied intervention. The FNR on the other hand describes the fraction of positive samples (mistakes of the CBM), which did not get a reapplied intervention. Naturally, the FNR is relatively large, as all mistakes of the CBM include systematic mistakes (*e.g.*, caused by data unbalance or confounders), which we want to mitigate with CB2M, as well as normal model mistakes, due to individual outliers, which CB2M is not designed to handle (see discussion above). Additionally, during setup CB2Ms where optimized more for precision rather than recall.

## A.7 FURTHER DETAILS ON MISTAKE DETECTION

In the experimental evaluation, we compared both CB2M and softmax for detecting model mistakes. These methods are however not exclusive, but could also be combined. In Tab. 12, we show the results of the mistake detection when combining both softmax and CB2M. We combined both methods either by full agreement, *i.e.*, only detect a mistake if both methods do so, or by partial-detection, *i.e.*, already detecting a mistake if only one of the methods does so. Selecting the exact strategy on the validation set enabled the combination of both methods to always perform as good as the previously better method, successfully combining both CB2M and softmax.

Table 12: Combination of CB2M and softmax for detection

| Dataset | Metric | Softmax | CB2M | Combined |
|---|---|---|---|---|
| CUB | AUROC (↑) | $83.7 \pm 1.1$ | $84.8 \pm 0.7$ | $85.0 \pm 0.5$ |
| | AUPR (↑) | $94.0 \pm 0.6$ | $94.6 \pm 0.3$ | $94.8 \pm 0.3$ |
| CUB (conf) | AUROC (↑) | $77.4 \pm 1.1$ | $85.1 \pm 0.5$ | $85.4 \pm 0.5$ |
| | AUPR (↑) | $91.5 \pm 0.7$ | $94.5 \pm 0.3$ | $94.7 \pm 0.3$ |
| Parity MNIST (unbalanced) | AUROC (↑) | $90.7 \pm 1.7$ | $88.7 \pm 0.4$ | $90.7 \pm 1.7$ |
| | AUPR (↑) | $98.8 \pm 0.3$ | $98.5 \pm 0.1$ | $98.8 \pm 0.3$ |
| Parity C-MNIST | AUROC (↑) | $65.7 \pm 0.3$ | $83.4 \pm 0.8$ | $83.6 \pm 0.5$ |
| | AUPR (↑) | $79.8 \pm 0.3$ | $91.5 \pm 0.4$ | $91.6 \pm 0.3$ |

| Exp | Dataset | $k$ | $t_d$ | $t_a$ |
|---|---|---|---|---|
| Tab 1 | CUB (a) | 1 | 3.5 | - |
| | Parity MNIST (ub) | 1 | $5, 5, 4, 4, 4$ | - |
| | Parity CMNIST | 1 | $7.5, 8.0, 7.0, 7.5, 8.5$ | - |
| Tab 2;3 | CUB | $3, 2, 3, 2, 4$ | $10, 11, 10, 10, 11$ | $0.99, 0.97, 0.99, 0.99, 0.99$ |
| | CUB (conf) | $1, 5, 4, 5, 3$ | $12, 12, 12, 11, 12$ | $0.99, 0.98, 0.98, 0.99, 0.97$ |
| | Parity MNIST (ub) | $2, 2, 1, 1, 1$ | $6, 6, 6, 5, 6$ | $0.99, 0.99, 0.98, 0.99, 0.99$ |
| | Parity CMNIST | $1, 3, 4, 3, 2$ | $3, 3, 4, 3, 3$ | $0.98, 0.99, 0.99, 0.97, 0.99$ |

Table 13: Used hyperparameters for all combinations of experiment and dataset. Cells contain values for all 5 seeds (except for CUB (a) where we have the same hyperparameter setting for all augmentations.

## A.8 HYPERPARAMETERS

To get values for the hyperparameters of CB2M, we performed a straightforward grid-based hyper-parameter optimization for $t_d$, $t_a$ and $k$, using training and validation set. For the selection of the distance threshold, we first computed the average distance of encodings from the validation set to have a suitable starting point for $t_d$. As the evaluation of a hyperparameter setting for CB2M does not entail any model training, the evaluation of different hyperparameter sets is computationally inexpensive. The detailed hyperparameter for each setup can be found in Tab. 13. For further training setup, *e.g.*, learning rates, we refer to the code.

