# OpenReview forum: "Learning to Intervene on Concept Bottlenecks"
_ICLR.cc/2024/Conference — Submitted to ICLR 2024_

### Official Review · Reviewer_XnGw · 2023-10-23

**Soundness:** 2 fair
**Presentation:** 3 good
**Contribution:** 4 excellent
**Rating:** 6
**Confidence:** 4

**Summary:**

Concept Bottleneck Models (CBMs) are an increasingly popular model class, designed to be more interpretable – and importantly *intervenable* by human users. However, querying humans for interventions on these models can be expensive, and models may make the same mistake repeatedly, resulting in repetitive interventions needed by users. In this work, these authors call attention to this problem and propose a new method – CB2M – to reconcile lack of reuse of interventions. CB2M is a modular extension to CBMs which leverages two memory banks: one which helps the model identify when a mistake is likely made in the output, and a second which reuses past interventions to correct such a mistake. The authors demonstrate the potential utility of CB2Ms across a range of experiments.

**Strengths:**

The motivation for the work is superb. The authors call attention to an incredibly important and under-recognized problem in CBMs: that models may make the same mistake repeatedly, and requiring humans to inculcate the same intervention over-and-over again can be cognitively demanding – and ought to be unnecessary. The authors’ proposed method is clever and has the potential to have great impact in the broader CBM community. I believe the authors offer value to the ICLR – and broader human-centric ML –  communities by 1) calling attention to this reuse problem, and 2) offering a first possible solution. The paper is also very well-written.

**Weaknesses:**

While I believe that simply calling attention to the reuse problem in interventions, coupled with their method proposal, holds value for the broader community – I do not think that the experiments in their current form sufficiently demonstrate the value of CB2M. Experimental validity therefore holds me back from assigning a higher overall score. I believe sizable further experiments are needed to really strengthen the work (or at least clarification on the current interpretation).

First, I am confused as to why the performance in Table 1 is lower for CB2Ms in the Full versus the Identified sets. Are examples in the Identified set those in which the memory module predicts that the example is misclassified? If so, why is the baseline CBM task performance so high (this would imply a high false positive rate?) It would be good for the authors to expand on possible False Positive and False Negative rates of the memory module, for each domain.

Second, the authors do not discuss the impact of the size of the memory module on performance. It would be good for the authors to have some kind of experiment(s) looking into the impact of thresholded sizes on the memory and intervention modules, as in practice, it’s possible that we may not be able to store all past instances?

Third, I am not convinced that the authors’ selection of baselines is adequate. As the authors detail in the Related Work, there have been several efforts to learn intervention policies (e.g., CooP) which select which next example to query people over. Yet, the authors never compare to any of these policies. None of these prior policies, to my understanding, leverage reuse of past interventions – as such, the authors could make the case that their method is complementary to these approaches; i.e., could be combined with methods which learn to intervene, which could justify not including such a baseline. However, the authors do not make such claims. I would be keen for the authors in the rebuttal to expand on how their work relates to other intervention policies and why they did not compare against them as baselines.

Fourth, the authors emphasize that their approach is model-agnostic. However, all experiments are with a CBM backbone. In the absence of augmenting other concept-based systems with their modules, I do not think the authors should claim their method is model-agnostic. If the authors would like to emphasize this claim, I believe experiments are needed with at least one other concept-based system. Otherwise, I think it is fine to leave for future work, but the text should be couched as such.

Lastly, I do not think the authors adequately discuss the limitations of their work (see Questions below). It would be good for there to be a dedicated Limitations section, or at least further prose on the matter.

**Questions:**

I have raised most of my important questions in the Weaknesses section. In addition:

- I am confused and concerned by Table 9 in the Appendix. The standard deviation is massive for CUB in particular. The authors note that the wide variance could be due to the threshold selection. However, it’s not clear to me why the threshold selected across the 5 seeds would vary so much that it leads to this level of variance in the number of generalized interventions? What is the variance in the selected threshold (can you please provide example threshold values?) and/or further explanation of what is happening here?
- It’s not clear to me that CB2M is better than softmax at detecting (with the exception of the Parity C-MNIST domain). The performance of CB2M and softmax are within error bounds for CUB. Can the authors expand on why this may be further?
- The authors’ assumption of perfect humans is sensible for this work. However, I would encourage the authors to think about how their module(s) may be challenged if human interventions are incorrect (or uncertain – e.g., Collins, Espinosa-Zarlenga et al, “Human Uncertainty in Concept Based Systems” AIES 2023). Such challenges would be worth expanding on in a Limitations section (of which the authors do not suitably have here).
- As a minor sematic note (which does not affect my score): the authors caption Fig 3 as “Less is more” — but really, this shows that the less is enough / sufficient to achieve high task performance; not that less is more than having further interventions. I’d encourage the authors to change that caption :)
- As another note, which does not impact my score, but would be nice for a revised version: have the authors looked at qualitative examples of the interventions / mistakes captured in the module? (e.g., Fig 2 of Chauhan et al, 2022)

---

> ### Author Response · Authors · 2023-11-17
>
> We thank you for your time and the valuable review. We are delighted to hear that you find the motivation of our work "superb" and the paper well-written.
> In the following, we address the concerns and questions posed in your review.
>
> - **Why is the Performance of CB2M in Tab 1 worse on full set compared to the identified instances?** With identified instances we consider all instances of the validation set to which the CB2M reapplies an intervention. Thus, the identified instances row shows that reapplying an intervention with CB2M does in general result in a good concept and class accuracy on these examples (after reapplying). Thus, the full set includes the samples of this identified set, but also all remaining (mistake) samples. This is the reason for the perceived lower accuracy on the full vs identified set. We have restructured Tab. 1 to make this more clear. Indeed, the concept accuracy of the base CBM on the identified instances seems comparably large. A few concept errors can, however, already result in a large drop in task performance, as observed in class performance of CBM on the identified set. Nevertheless, we have provided explicit FPR and FNR for the generalization task in the appendix (App. A.6). These suggest quite low FPR for CB2Ms, in contrast to a high FNR. However we assume this to be due to hyperparameter tuning for high precision and the fact that negative samples consist of all base model errors, including those that are in principle not correctable via generalizing interventions.
> - **Comparison to baselines (CooP und Shin et al.)**:
> In fact, the goal of our work is not to propose a method that stands in contrast to the mentioned works (CooP and Shin et al.), but rather CB2M represents a method that can be considered orthogonal to these. Specifically, Shin et al. propose several heuristics to estimate which concepts are the most important to intervene on for a given sample. Thus, they do not provide a method to estimate which samples should receive interventions, but instead “only” provide a concept ordering for given samples. Similarly, CooP can provide a subset of concepts to intervene on (even over multiple samples). However, CooP cannot be used directly to identify whether a sample in isolation should receive an intervention. Moreover, neither of these methods tackle the problem of the inefficiency of one-time interventions. In contrast, CB2M can be used to detect samples which potentially require interventions and can then even generalize given interventions to new samples. Thus, overall the mentioned approaches tackle different problem settings such that CB2Ms do not stand in opposition to these, but can in principle be used in conjunction with them (as e.g., Fig. 3 suggests).
> - **CB2M is not model-agnostic**:
> Indeed, our approach is not in general model-agnostic, but rather agnostic in the context of CBM-like architectures, as long as they have an accessible concept layer and encodings. We have clarified this in the paper and updated the wording where necessary
> - **High standard deviation in Tab 9 in the appendix**:
> This happens due to two facts: We optimized $t_d$ only once for all augmentations, which is not necessarily optimal given the individual nature of each augmentation. Additionally, the augmentations salt&pepper and speckles have a more disruptive effect on the model encodings, causing fewer samples to be recognized for interventions. We have added a more detailed discussion on this in App. A.6.
> - **Detection of mistakes for softmax and CB2M is not different (except for C-MNIST)**:
> Indeed, our results do suggest the baseline softmax approach to be en par with CB2M. However a major difference between softmax and CB2M is that the softmax approach looks at the model in total. Thus it can, in principle, not distinguish between bottleneck and predictor mistakes. In our investigated data set settings prediction errors are solely caused by bottleneck errors. In more natural and complex settings in which both types of errors occur (bottleneck and predictor error) using the softmax approach will be insufficient. Moreover, softmax and CB2M are not exclusive, as we show in App. A.7 where we combine both approaches into one.
> - **Assumption of perfect human interventions**:
> Yes, indeed this is an assumption of our work. We have added a discussion on this in a new limitations section (at the end of the  evaluations section), linking the mentioned reference as well as to other relevant works.
> - **Less is more, Fig. 3**:
> Thanks for pointing that out, we have modified the caption.
>
> We hope that this clarifies your remaining concerns. Otherwise, we are happy to answer any remaining questions.

---

> > ### Comment · Reviewer_XnGw · 2023-11-22
> > **Thank you for your response!**
> >
> > Dear Authors,
> >
> > I admire your thorough response! Thank you for all the work you have done to update the submission. I think your Limitations section is particular useful and appreciate the clarifications in the Appendix. I have raised my score accordingly.
> >
> > However, I do resonate with Reviewer fRVj's concerns. I think Points 1 and 2 of Reviewer fRVj's recent response are particularly important -- if not essential -- to include in the Limitations or main text somewhere.
> >
> > I do think that this paper has a very strong motivation, as I noted, and think it can inspire further work in the burgeoning concept-based AI system community, hence my increased score. However, I do strongly urge the authors to consider further updates to the text per the above.

---

> > > ### Author Response · Authors · 2023-11-22
> > >
> > > Thank you for your time and appreciation of our work and updates. In the context of referring to reviewer fRVj's additional remarks, we have updated the paper accordingly, specifically focusing on fRVj's initial concerns 1 and 2 (cf. our latest response to them). We are happy if this reviewer checks these updates and reconsiders their score again.

---

### Official Review · Reviewer_oJVX · 2023-10-30

**Soundness:** 3 good
**Presentation:** 3 good
**Contribution:** 2 fair
**Rating:** 6
**Confidence:** 4

**Summary:**

The paper proposes an extension to CBM architectures called CB2M in which interventions are not just used once, but rather stored in memory and reused on test data to improve performance and detect possibly similar errors. Results show their method is better than normal CBMs at doing this.

**Strengths:**

The paper has an interesting idea, I like the notion of learning from human feedback and improving over time or fixing edge cases in which the neural network is making the same mistakes over and over again.

I also like the possibility of human-AI collaboration here where we could e.g. detect errors humans and/or AI are making at test time to try and make up the difference between the two. I certainly think this direction has a lot of potential and will form an important part of explainable ML going forward.

Essentially this all comes from storing these cases in memory, but it's worth mentioning that this isn't the most novel idea (a lot so similar work exists in CBR etc.), but in this context of CBMs it is reasonably interesting.

**Weaknesses:**

The weakness of the paper is the evaluation I feel. The authors setup a few situations on a few common datasets to show the utility of their method. However, none of these experimental setups are particularly compelling, and somewhat contrived. I'm not sure if the point of the method is to increase model performance, or HCI, etc...

From a performance perspective, take the CUB dataset, SOTA on this dataset is (last I checked) 92%+, but here their method is 88.7%. I understand that raw performance is probably not your goal here, but if you're using accuracy to assess the usefulness of your method, then this isn't really very compelling, as I can't e.g. use CB2M to squeeze more accuracy out of my models, so I am left wondering how it would be useful there. What would be great is to show you could break this 92% ceiling with your method, human feedback, and interventions etc...

In another vein, the human interventions are simulated, which again makes me wonder if humans could actually interact with the method how the authors propose they could.

If the authors could show their method e.g. working with a doctor to improve team performance overall, or just improve accuracy over a standard black-box, that would be very exciting, but they don't. So, I am left wondering what the application of this is at all.

### Small things
* The first two figures don't explain $f$ or $g$, the figures should stand alone usually.
* Page 2: this issue (2) by... should be -- this issue by (2)....
* I would tend to axe the third contribution on page 2, it's just experiment results which is expected.
* It's not clear where $x_e$ is taken.
* Second paragraph on page 4 would probably help the intro motivation.
* $t_d$ needs to be clearly explained how the value was taken (unless I missed it sorry)
* Eq 2: I wouldn't use $val$, it reminds me of "validation" personally, which is confusing.

**Questions:**

* What is a real-world application of this method that could make people in the ML community genuinely excited? Something were the method could be shown to be *understandable* and *useful* to intended practitioners of the system.
* See above for my other general critiques.

Overall, I like the paper's core idea, and I veer slightly (although just slightly) towards acceptance, but I will mutate this after the next phase depending on my interactions with the AC, reviewers, and authors.

---

> ### Author Response · Authors · 2023-11-17
>
> Thank you for reviewing our paper, we are happy to hear that you like our approach and provided valuable feedback.
>
> We hope we can answer your questions in the following:
>
> - **What is the concrete goal and potential use-cases of your method?** Overall, in this work, we propose to extend CBMs with an inspectable and interactable memory module. This, among over things allows to increase the effectiveness of user interactions on CBMs in
> two major ways: First, a CB2M is able to propose a preselection of potential errors, which can then be presented to a human user. By effectively guiding the users attention to model mistakes, this preselection potentially reduces the amount of required human inspection
> and interactions. Second, given the generalization abilities of the CB2M’s memory module, interventional feedback on few samples can effectively be transferred to novel and “unintervened” samples. Thus, overall the goal of CB2Ms is to more efficiently and effectively integrate interventional feedback from expert human users into concept bottleneck models. In fact, as the reviewer hints, the medical domain is a very valid use-case of CB2Ms. Particularly, such high-stake decision domains require extensive and detailed human expert inspection, cf. [1]. However, as high-quality feedback from medical doctors is generally expensive and sparse in nature, it is specifically necessary to make the most of those interactions that can be provided. This is particularly, where CB2Ms can shine. We have demonstrated these abilities on several benchmark data sets, among other things confounded data sets. Specifically, in our updated version we have added a second confounded data set (CUB (conf.)) to make these points more clear.
>
> - **CB2M does not achieve SOTA results on CUB**: From a performance perspective, the objective of CB2M is not necessarily to achieve new SOTA results, but CB2Ms are rather specifically designed as a flexible extension to increase the effectiveness of human inter-
> actions (see above) on any CBM-like base model ($g \circ f$). In this work we introduce them in the context of vanilla CBMs based on Koh et al. (2020)’s work. Nevertheless, they can in principle also be combined with more recent architectures which provide a higher baseline performance as the ones which the reviewer suggest. Moreover, the results presented in Tab. 1 in fact depict model performances based on the augmented CUB version, thus explaining the differences which the reviewer hints at this point. We have now marked this as (Aug.) in the paper to make this more clear.
>
> - **It is not clear where $x_e$ is taken**:
> In fact, the encoding $x_e$ for a sample $x$ is the input of the last layer of the bottleneck network $g$ (Sec. 2.2, first paragraph).
> - **Confusion concerning val notation**: Indeed, we have updated this to valid(·).
>
> We hope that we have clarified your concerns. Do let us know if you have any further questions or comments.
>
> [1] DeGrave, A. J., Janizek, J. D., & Lee, S. I. (2021). AI for radiographic COVID-19 detection selects shortcuts over signal. Nature Machine Intelligence, 3(7), 610-619.

---

> > ### Comment · Reviewer_oJVX · 2023-11-22
> >
> > Thanks a lot for your feedback! I think you are on the right track.
> >
> > I have read everything and considered carefully. I agree with almost everything you say, I think though that we part ways on the evaluation part, which was my main concern.
> >
> > I think that XAI has a multitude of methods published, but almost never are they tested in a realistic deployment setting to prove they are understandable and useful to intended practitioners of the system.
> >
> > I like your paper, which is why I accepted it, but I would need to see a strong user evaluation showing its utility in practice to raise my score more. But again I concede that this is not typical of ICLR, which is why I still veered towards acceptance, I wish you good luck with this research!

---

> > > ### Author Response · Authors · 2023-11-22
> > >
> > > Thanks so much for taking the time to go through our changes and answers as well as appreciating the work with acceptance. We share the reviewers concerns related to XAI research and claims often made therein. Indeed, whether CBMs represent more understandable models is an important open topic in the community, yet orthogonal to our work. More importantly, we agree that the utility of interventions as introduced in the original CBM paper (Koh et al. (2020)) could benefit from more user evaluations. While this is a pressing issue, it is beyond the scope of our work. Specifically, given the assumption that interventions in general are useful for users, CB2Ms improve the efficiency of these interventions once they are provided. Again we very much appreciate the fruitful discussion and are happy if you reconsider your score.

---

### Official Review · Reviewer_fRVj · 2023-11-01

**Soundness:** 1 poor
**Presentation:** 3 good
**Contribution:** 2 fair
**Rating:** 5
**Confidence:** 4

**Summary:**

This paper proposes Concept Bottleneck Memory Models (CB2Ms), a new model-agnostic extension of Concept Bottleneck Models (CBMs) in which an adaptive memory is incorporated to improve a CBM’s receptiveness to test-time interventions and its uptake of feedback at test-time. Through two sets of distance-based memory banks, namely an *intervention memory* and a *mistake memory*,  CB2Ms learn to identify potentially mispredicted samples and reapply previous interventions to automatically improve the concept and task accuracy after only a handful of test-time interventions have been performed. This work evaluates CB2Ms on four datasets (two MNIST-based datasets and two real-world datasets) and shows that the proposed extensions enable CBM-based models to significantly boost their intervention performance and their ability to be more robust to distribution shift and train-time spurious correlations.

**Strengths:**

Thank you for submitting this work! I believe this is a very interesting idea and something that has certainly not been carefully explored in the concept-based literature before. Explicitly, I believe the following are the main strengths of this paper:

1. **Originality**: the idea of incorporating a test-time adaptive memory to improve the uptake of intervention feedback and avoid discarding potentially useful information provided at intervention-time, is certainly original in the field of concept-based explainability. Furthermore, the work is well-placed within this area with a set of diverse related works discussed in this paper and potential ideas mentioned in the end discussion.
2. **Quality**: although I believe the experimental set-up could have benefited from a more careful design (see below), the quality of the presented idea, as the presentation of the idea itself, is up to the standards of work in this conference and community.
3. **Clarity**: the paper’s writing is very easy to follow, with almost no typos and a structure which makes the reading flow easily from beginning to end. This helps the authors clearly communicate their ideas and the motivation behind the ideas. Furthermore, the inclusion of a code base helps to understand the clarity of this work and promotes reproducibility, both highly desirable features.
4. **Significance**: the paper’s main contribution, that of a model-agnostic mechanism to take test-time feedback into account when considering future interventions, is certainly significant in the concept-based literature and may lead to further advancement in the near future. Nevertheless, this significance is contingent on a careful and fair evaluation of the proposed method (see weaknesses below). If the doubts I have regarding this paper's experiments are carefully discussed/corrected, and the paper’s main claims still hold, then I believe this paper would be of good value to the community.

**Weaknesses:**

Although I think this paper has several strengths, as outlined above, I am concerned about the fairness of its evaluation and the lack of baselines that would be relevant comparison methods with what is presented in this manuscript. In particular, I believe these are the paper's main weaknesses:
1. [Critical] My biggest concern with this paper is its evaluation. In my opinion, it is not fair to evaluate a method like CB2M, which can take and store extra samples at test time to improve performance on future unseen samples, against methods that completely ignore the same feedback even if in theory it could be used to improve their performance as well. For example, given that the validation set is used to construct the initial mistake memory, at the very least I would expect to see as a fair baseline a CBM that was able to update its weights using feedback from the same validation set during training (as otherwise CB2M is unfairly being trained on more data than the CBM baseline!). Similarly, when considering how the intervention memory is used, it would be fair to have as a baseline a CBM whose weights are updated at test time so that feedback from a new intervention is considered in the model. This can be done, say, via a variety of online learning algorithms that aim to minimize the cross entropy loss of the corrected concept prediction given the provided ground-truth (intervened) label (or even a baseline that runs a few gradient decent steps on this new intervention's label and corresponding sample). Having such a baseline and showing the CB2M still beats that baseline would provide very strong evidence that the memory mechanism introduced in this paper is worthwhile and novel compared to methods that already exist there to address similar problems.
1. [Critical] Regarding the evaluation of the method on unseen data points (e.g., Table 1), it is unclear whether it is fair for ECB2Ms to be able to use the entire (unmodified) test set as part of their memory loading while none of the baselines can have the same benefit at training time. Evaluating CB2M and other baselines on a modified version of this test set seems like a very unfair comparison if none of the other methods were able to obtain any feedback from simulated interventions on the unmodified version of these samples. It may be fairer to avoid any sort of leakage from the test set into the training set of CB2M by avoiding loading into its initial memory samples that are highly related to the ones it will be evaluated on (this can be done by splitting the test set into two). I understand this is how this method may be used in practice (with the memory taking advantage of test-time samples to build a database of interventions and mistakes),; comparing it against a CBM's results on the same table makes it seem like this is an apples-with-apples comparison when in reality it may not be such. Making this distinction clearer, and if possible the evaluation fairer, would help a lot to indicate how this method works and how it is expected to be used.
1. [Critical] With the exception of C-MNIST, the results shown in the identification of mistaken samples do not appear to be statistically significant (see the standard deviations). Because of this, it is unclear whether the proposed method identifies errors better than, say, the naive softmax baseline. Similarly, the effect on generalized interventions based on CB2M's detection on the full dataset (Table 3) does not appear to be statistically significantly better than what one observed when using the Softmax detection for all datasets (see standard deviations). This casts some doubt on the usability of this method.
1. [Major] The use of a dynamic memory in CB2M implies that this method will either (a) struggle to scale to large concept spaces or spaces with a lot of variability in concepts (as it will require a significant number of examples before capturing the variance of the concept space and this requires CB2M to store all these sample's embeddings), or (b) require one will have to cap the size of the memory, leading to another hyperparameter that needs fine-tuning. As discussed in the questions below, the size of the memory may also lead to intractability at test time due to a large search space needed when correcting a mistake that was identified via the mistake memory. These aspects are not discussed anywhere in this paper.
1. [Major] The proposed method depends on three hyperparameters that are reportedly crucial for the end performance of the model, namely $t_d$, $t_a$, and $k$. Nevertheless, I could not find a reference anywhere in this paper on how these hyperparameters are selected for the experiments reported. Notice that the mechanism to fine-tune $t_d$ based on a validation set is explained near Equations 2 and 3. However,  the actual values used to perform this validation-based search are not reported anywhere for the experiments in this paper. Furthermore, the dependency of $t_d$ with $t_a$ and $k$ is also not elaborated anywhere in the main text (although it is understood that $k = 1$ when doing generalized interventions).
1. [Major] Related to the point above, there are no ablations showing how sensitive the results reported in this paper are to these hyperparameters and how important the validation set is to fine-tune them correctly. This hinders the understanding of how this method would fare in practice and how easy it is to use.
1. [Major] It is unclear how the size of the validation set affects any of the results observed. Similarly, it is unclear how the number of test-time interventions affects the results seen (a crucial element to understand given the role these interventions take in improving the method's future test-time performance). More importantly, no ablations are provided to answer these important questions.

**Questions:**

Given my concerns outlined in the weaknesses above, I am leaning towards rejection at the moment. Nevertheless, I am more than happy to be proven wrong or corrected if I misunderstood a crucial part of this work. With this in mind, I hope the following questions, in no particular order, would help clarify some of the doubts on this work. If possible, I would appreciate it if the authors could elaborate on these concerns as they may serve as a good starting point for a discussion during rebuttal:
1. Regarding my concern about the fairness of the evaluation, could you please let me know if I am misunderstanding something here? If not, could you please elaborate on how CB2Ms would fare against similar (fairer) baselines as the ones discussed in the weaknesses?
1. Regarding my concern on the results of Table 1: could you please elaborate on why it would not be more fair to perform the evaluation on a dataset of samples whose unmodified versions have not been used to set up the initial memory of CB2M?
1. Regarding my comments on the weaknesses for Figure 3: how would the curve shown for CB2M look vis-a-vis that of a vanilla CBM in which multiple groups are randomly intervened on? I am trying to fully understand how these results are unexpected or different to those seen on CBMs and CEMs in previous works (e.g., Shin et al. and Chauhan et al., both works cited on this paper).
1. Could you please elaborate on the importance of the validation set size and the number of test-time interventions before evaluation on the results presented in this paper?
1. Could you please elaborate on the hyperparameter selection process for the experiments in this paper (see weakness above)?
1. Do you have a sense of how sensitive CB2Ms are to their hyperparameters? Are there any good strategies to select these hyperparameters?
1. Similar to the question above, could you elaborate on how important the memory size is for this model? Are there any ablations on how memory size affects the results? It is unreasonable to assume that one can have a boundless memory for CB2M; therefore, fully understanding this question is essential.
1. Similarly, how is the inference wall-clock performance affected by introducing the memory banks at inference time? I imagine there will be a hit but fully understanding how significant this hit is before this method becomes intractable is absolutely crucial for understanding its weaknesses and strengths.
1. Just to confirm: is it the case that during the evaluation of CB2M the memory is left unmodified once evaluation starts on the test set? I would expect that to be the case for this evaluation to be fair, however I could not easily find this detail in the paper.
1. For the results of Table 4, how many test interventions are needed for CB2M to achieve the observed results? I could not find this detail easily.

Besides these questions, I found the following typos/minor errors that, if fixed, could improve the quality of the current manuscript:
1. Page 4: "Thus, with the ability to handle task (i)..." seems to be missing something before the enumeration "(i)" begins.
1. Page 8: "improve a model via on the detected mistakes" should probably be "improve a model via the detected mistakes"
1. Page 8: Missing space in "improvements.Even"
1. nit on Page 8: closing quotation is used for the beginning of "full" instead of the opening quotation (`` in LaTeX)
1. Page 8: period used instead of comma in "...the distribution shift. indicating..."

---

> ### Author Response · Authors · 2023-11-17
>
> We thank you for the extensive review, pointing out valuable strengths of our paper as well as great suggestions for improvement. In the following, we addressed your questions:
>
> - **Question 1 & 2: On the fairness of evaluation between CBM and CB2M**: Indeed, CB2M has access to more information than CBM in the evaluation. We provide results for CBM finetuned on the respective validation sets (or the test set in case of CUB (Aug.)) in our
> updated version (App. A.6) where we differentiate between different amounts of finetuning steps (i.e. finetuning epochs). We observe that in all cases, CB2M performs better than doing few finetuning steps, however, for CUB and Parity CMNIST, finetuning for an extended amount of time outperforms CB2M. On the other side, finetuning on Parity-MNIST (unbalanced) does not provide any real benefit, even for an extended number of steps.
>
>   Though we agree to a certain degree with the reviewer on comparing the models with the same amount of data, these comparisons are also unfair towards CB2Ms in the context that CB2M does not perform parameter optimization based on the validation set, whereas the CBM’s parameters in Tab. 10 were specifically finetuned on this. Additionally, a particular benefit of CB2M (in comparison to CBM  inetuning) is the potential use-case of dynamic, online learning settings during model deployment. In this case, finetuning on each novel set of data is infeasible and might even lead to issues such as catastrophic forgetting.
>
>   Moreover, this form of finetuning the CBM’s parameters on the interventional data has even more drawbacks in comparison to CB2M in the context of interpretablity and interactability. Specifically, it is very difficult to remove already applied interventions from the finetuned model, if it turns out these interventions were bad/incorrect. On the other hand it is difficult to inspect the representation of the finetuned model, where in CB2M a user can simply inspect the model’s memory.
>
> - **Question 3: Comparison of Fig.3 with a CBM**: This figure should rather highlight that findings of the related literature (concrete from Shin et al. in this case) also hold when using CB2M. In other words, Fig.3 shows that it is not necessary to provide CB2M with interventions to all concepts to receive significantly improved class accuracies.
>
> - **Question 9: Memory of CB2M is not changed during evaluation?** Yes, this is correct.
> - **Question 10: How many interventions have been used for tab 4 (distribution shift)?** We used a validation split as for MNIST, consisting of 20% of the training data of SVHN (14000 images). We have added this information to the paper.
>
> We hope that the remaining questions (4 to 8) have been answered within the general remarks. If that should not be the case, we will be happy to provide further clarifications.

---

> > ### Comment · Reviewer_fRVj · 2023-11-22
> >
> > Dear Authors,
> >
> > Thank you so much for your thoughtful rebuttal and answers to my questions/concerns. I really appreciate all your results and new additions regarding hyperparameters, memory size, and the effect of the validation set. These additions answered my questions on these matters.
> >
> > Regarding my concern on the evaluation of this method, which is arguably my biggest concern as discussed in my review, I appreciate the inclusion of Table 10 in Appendix A.6. Nevertheless, I have to admit that my concern on this issue has not been fully assuaged after going over the new additions and the rebuttal. The reasons for this are the following:
> > 1. The fine-tuned and vanilla online learning baselines I suggested are, in my opinion, a very basic and fair set of baselines. Therefore, I believe they should probably be included as part of the main experimental section rather than in an appendix. At the minimum, I would suggest that the main takeaways of the appendix are summarized within the main body of the paper.
> > 2. The fact that fine-tuning a CBM with the validation set for 5-10 more epochs can improve performance compared to having the memory of CB2M (and also results in significantly smaller standard deviations) is noteworthy and casts some doubt on the reliability of the proposed method. Regarding the authors' comment that this is an unfair evaluation against CB2M, I would disagree with this. My disagreement comes from the fact that looking over a memory during inference has accompanying costs that are otherwise amortized if one instead proceeds with the fine-tuning approach (i.e., computational costs incurred during fine-tuning are not too different from costs incurred by CB2M at inference time when searching over the memory). Furthermore, the fine-tuning approach comes with extra memory requirements that do not appear in the fine-tuning setup.
> > 3. Regarding the benefit of CB2M enabling more efficient online learning than fine-tuning, I agree with the authors when one considers only vanilla fine-tuning (the most basic online learning baseline which would also be a fair evaluation baseline). Nevertheless, there are several online learning algorithms designed to improve efficiency and avoid catastrophic forgetting (see [1] for an example of a known continual learning approach to prevent catastrophic forgetting). All such methods are all in direct competition with this method, in particular when dealing with distribution shifts during inference, and are not included as baselines in the evaluation of this work. I would argue that it is ok not to include them all as baselines but it may be worth discussing them somewhere in the paper. Finally,  it is worth pointing out that it may be the case that catastrophic forgetting could occur in CB2M in the limit of a growing number of saved test interventions (I can't see why this could not be the case in particular when a distribution shift occurs). More evidence would be required to understand if CB2M is not prone to this phenomenon.
> > 4. I like the suggestion that CB2M may be better than fine-tuning at removing potentially mistaken interventions! This is particularly interesting, especially if one can identify these interventions. However, under the setup used for this paper, where interventions are assumed to all be correct, and where the validation dataset is assumed to have valid concept and task label annotations for all samples, this is not quite a disadvantage of simply fine-tuning the model for a few epochs.
> > 5. I agree with your assessment that analyzing a memory bank may be easier than analyzing latent representations. In this case, I do think CB2M's memory has an advantage over fine-tuning.
> >
> > Finally, I was wondering if I missed something, but do you have any comments on my concern about the test-train leakage I mentioned in my second critical weakness and question?  I could not find any specific answers to this question/concern in the rebuttal or updated paper, but I might've missed it (sorry if I did!).
> >
> > These concerns, together with my comments on the lack of statistical significance of some of the experiments (e.g., Table 3), make me continue to lean towards rejection rather than acceptance. However, given the authors' replies to some of my concerns and their proper addressing of our questions on hyperparameters, I will increase my score to a borderline reject. If the authors or other reviewers have a strong case against the concerns detailed in this discussion, I am willing to change my score if such a case can be made without substantially more evidence on the part of the authors (e.g., needing significantly more large-scale experiments).
> >
> > I wish you the absolute best with this submission and your future research!
> >
> > ### References
> > [1] Aljundi, Rahaf, et al. "Gradient based sample selection for online continual learning." Advances in neural information processing systems 32 (2019).

---

> > > ### Author Response · Authors · 2023-11-22
> > >
> > > Thanks so much for your time and we very much appreciate the fruitful discussion. We have some points below:
> > >
> > >  - **1.** Yes, we have added the results of Tab. 10 to Tab. 1 as additional baselines and discuss the results in the main text.
> > >  - **2.**  We agree that it is an open question whether parameter finetuning is indeed overall more cost-efficient than adaptation via memory. It is plausible that in specific settings finetuning the model parameter may be a "better" solution. However, in other cases such as finetuning exceptionally large models (e.g. LLMs), the training cost can be higher than the memory costs. Together with our results that indicate that finetuning does not always yield better performances (cf. results on Parity MNIST (unbalanced)) and the potential of using very efficient memory retrieval, we thus propose to consider CB2M as an complementary approach for model revisions via interventions. We have added this point to Sec. 3.1.
> > >  - **3.** Yes we agree. As the reviewer suggests we have added this discussion to Sec. 3.1 and future work.
> > >  - **Train/Test leakage:** On CUB, we specifically want to evaluate how well CB2M can generalize interventions when similar versions of intervened data points reoccur (one of the motivations for CB2M). Therefore, this "leakage" between the unmodified and augmented versions of the test set is deliberate. We agree that the comparison to the base CBM (without) finetuning is unfair, similar to the other datasets.
> > >  As we have added the results for the finetuned CBM (which also has access to the unmodified test set), we believe that this comparison is then again comparable to the other two datasets.
> > >
> > >  - **Stastical significance of Tab. 2 and Tab. 3:** Sorry, we simply missed to answer this point in our initial response (also see discussion with reviewer XnGw).
> > >  Indeed, our results do suggest the baseline softmax approach to be on par with CB2M. However a major difference between softmax and CB2M is that the softmax approach looks at the model in total. Thus it can, in principle, not distinguish between bottleneck and predictor mistakes. In our investigated data set settings prediction errors are solely caused by bottleneck errors. In more natural and complex settings in which both types of errors occur (bottleneck and predictor error) using the softmax approach will be insufficient. Moreover, softmax and CB2M are not exclusive, as we show in App. A.7 where we combine both approaches into one.
> > >  Consequently, we further adapted the discussion of Tab. 3 to place the results in better context.
> > >
> > > In light of our new remarks we are happy if the reviewer reconsiders their score again.

---

### Author Response · Authors · 2023-11-17

We wish to thank all reviewers for their valuable time and feedback. We are delighted that
all reviewers agree on the importance and motivation of our work and find the work well placed,
written and structured. In the following we provide responses to overarching remarks and provide
detailed responses individually.

- **Selection of hyperparameters**: In general, for optimizing the hyperparameters, e.g. $t_d$, $t_a$
and $k$, we perform a simple grid-based hyperparameter optimization based on the training
and validation sets. Specifically, for the distance threshold $t_d$, we based the search around
the average distance between encodings on the validation set. Thus, this optimizations
in general represents a straightforward approach which importantly does not require any
model training. Rather it is based on evaluating the memory of a CB2M which comes at
only a low computational cost. Additionally, we provided the used hyperparameters in the
appendix (App. A.8).
- **Discussion of size and efficiency of the memory module**: We agree that the size of the
memory module (i.e. validation set in our setting) is indeed an important aspect of the
evaluations. We therefore provide novel results of the generalizaton experiment when using different-sized subsets of the validation set in the appendix (App. A.5). As expected
with larger memory size the CB2M’s performances increase. Particularly, concerning both
Parity-MNIST datasets we observe preferential performance boosts given already small
subsets of the validation set, i.e. 25% of the data already leads to improvements beyond the
baseline performances.

  Moreover, other works on similar memory modules, i.e. nearest neighbor or retrieval based inference show a high effectivity even in the context of large databases [1, 2]. Particularly, these have been shown to perform well on much larger datasets than the ones investigated in this work. Based on this we expect the memory size and efficiency to scale well also in the context of CB2Ms, particularly given the fact that having millions of previous interventions is not to be anticipated. Also techniques such as memory condensation (e.g. prototype representations) could be applied to improve efficiency further.

Lastly, we note that we have added additional results in Tab. 2 on a novel, confounded version of
CUB.

[1] Johnson, J., Douze, M., & J´egou, H. (2019). Billion-scale similarity search with gpus. IEEE
Transactions on Big Data, 7(3), 535-547.

[2] Borgeaud, S., Mensch, A., Hoffmann, J., Cai, T., Rutherford, E., Millican, K., ... & Sifre, L.
(2022, June). Improving language models by retrieving from trillions of tokens. In International
conference on machine learning (pp. 2206-2240). PMLR

---

### Author Response · Authors · 2023-11-22
**Thank You!**

Dear reviewers, we would really like to highlight our appreciation for your detailed feedback and the great quality of fruitful discussions. Thank you for this!

---

### Meta-Review · Area_Chair_WSL7 · 2023-12-10

**Metareview:**

**Summary**: This paper presents a new class of concept bottleneck models with a "memory component." The component can store human interventions, thus reducing the number of human interventions and identifying incorrectly predicted samples. The authors proposed. This paper develops a model-agnostic approach to build the component and evaluates its approach on four datasets. The results highlight some of the potential benefits of this solution, such as an improvement in performance and resilience to distribution shift and train-time spurious correlations.

**Strengths**:
- Topic: the paper considers an important class of problems that would be of interest to those in the field
- Framework: the proposed approach addresses two of the main "bottlenecks" in the use of concept bottleneck models.

**Weaknesses**:

- Assumptions: The proposed memory component only makes sense in a setting where (i) correcting incorrect concepts will improve validity; (ii) the mapping from concepts to outcomes is deterministic. These are critical assumptions that are required to ensure mistakes can be fixed and that corrections will improve accuracy. The paper states these assumptions explicitly -- but provides little guidance on when we could expect them and how the system would work when they fail.

**What is Missing**

One of the key concerns in this case is that the paper discusses these issues in ways that are hand-wavy - even when there are concrete guidelines and recommendations. For example, Assumption (i) (i.e., "validity") should require that we train the independent CBM. Likewise, Assumption (ii) should require that we predict an outcome that is deterministic (e.g., the image contains a bird) rather than an outcome that is probabilistic (e.g., the patient has pneumonia). At a minimum, the paper would state these facts in a longer discussion surrounding each assumption so that practitioners can determine whether a CB2M module would work for them.

**Justification For Why Not Higher Score:**

See review.

**Justification For Why Not Lower Score:**

N/A

---

### Decision · Program_Chairs · 2024-01-16

Reject